# Pre- and postsynaptically expressed spike-timing-dependent plasticity contribute differentially to neuronal learning

**Beatriz Eymi Pimentel Mizusaki**[1,2,3], **Sally Si Ying Li**[1¤], **Rui Ponte Costa**[3,4,5], **Per Jesper Sjöström**[1] *

**1** Centre for Research in Neuroscience, Brain Repair and Integrative Neuroscience Programme, Departments of Medicine, Neurology and Neurosurgery, The Research Institute of the McGill University Health Centre, Montreal General Hospital, Montreal, Quebec, Canada, **2** Instituto de Física, Universidade Federal do Rio Grande do Sul, Porto Alegre, Rio Grande do Sul, Brazil, **3** Computational Neuroscience Unit, Department of Computer Science, SCEEM, Faculty of Engineering, University of Bristol, Bristol, United Kingdom, **4** Department of Physiology, University of Bern, Bern, Switzerland, **5** Centre for Neural Circuits and Behaviour, Department of Physiology, Anatomy and Genetics, University of Oxford, Oxford, United Kingdom

¤ Current address: The Solomon H. Snyder Department of Neuroscience, Johns Hopkins University, Baltimore, MD 21205, United States of America
* jesper.sjostrom@mcgill.ca

**Data Availability Statement:** Data set from Costa et al eLife 2015 is available at the Dryad Digital Repository, http://dx.doi.org/10.5061/dryad.p286g. The biologically tuned computer model is available at ModelDB, Accession: 184487. Computer models

## Abstract

A plethora of experimental studies have shown that long-term synaptic plasticity can be expressed pre- or postsynaptically depending on a range of factors such as developmental stage, synapse type, and activity patterns. The functional consequences of this diversity are not clear, although it is understood that whereas postsynaptic expression of plasticity predominantly affects synaptic response amplitude, presynaptic expression alters both synaptic response amplitude and short-term dynamics. In most models of neuronal learning, long-term synaptic plasticity is implemented as changes in connective weights. The consideration of long-term plasticity as a fixed change in amplitude corresponds more closely to post- than to presynaptic expression, which means theoretical outcomes based on this choice of implementation may have a postsynaptic bias. To explore the functional implications of the diversity of expression of long-term synaptic plasticity, we adapted a model of long-term plasticity, more specifically spike-timing-dependent plasticity (STDP), such that it was expressed either independently pre- or postsynaptically, or in a mixture of both ways. We compared pair-based standard STDP models and a biologically tuned triplet STDP model, and investigated the outcomes in a minimal setting, using two different learning schemes: in the first, inputs were triggered at different latencies, and in the second a subset of inputs were temporally correlated. We found that presynaptic changes adjusted the speed of learning, while postsynaptic expression was more efficient at regulating spike timing and frequency. When combining both expression loci, postsynaptic changes amplified the response range, while presynaptic plasticity allowed control over postsynaptic firing rates, potentially providing a form of activity homeostasis. Our findings highlight how the seemingly innocuous choice of implementing synaptic plasticity by single weight modification may unwittingly introduce a postsynaptic bias in modelling outcomes. We conclude that pre- and

introduced in this study are available at https://github.com/BMizusaki/pre-post_plasticity.

**Funding:** This work was funded by CNPq 202183/2015-7 (BEPM), Canada Summer Jobs (SSYL), EPSRC EP/F500385/1 (RPC), BBSRC BB/F529254/1 (RPC), Fundacao para a Ciencia e a Tecnologia SFRH/BD/60301/2009 (RPC), CFI LOF 28331 (PJS), CIHR OG 126137 (PJS), CIHR PG 156223 (PJS), CIHR NIA 288936 (PJS), FRQS CB Sr 254033 (PJS), NSERC DG 418546-2 (PJS), NSERC DG 2017-04730 (PJS), and NSERC DAS 2017-507818 (PJS). The funders played no role in the study design, data collection and analysis, decision to publish, or preparation of the manuscript.

**Competing interests:** The authors have declared that no competing interests exist.

postsynaptically expressed plasticity are not interchangeable, but enable complimentary functions.

## Author summary

Differences between functional properties of pre- or postsynaptically expressed long-term plasticity have not yet been explored in much detail. In this paper, we used minimalist models of STDP with different expression loci, in search of fundamental functional consequences. Biologically, presynaptic expression acts mostly on neurotransmitter release, thereby altering short-term synaptic dynamics, whereas postsynaptic expression affects mainly synaptic gain. We compared models where plasticity was expressed only presynaptically or postsynaptically, or in both ways. We found that postsynaptic plasticity had a bigger impact over response times, while both pre- and postsynaptic plasticity were similarly capable of detecting correlated inputs. A model with biologically tuned expression of plasticity achieved the same outcome over a range of frequencies. Also, postsynaptic spiking frequency was not directly affected by presynaptic plasticity of short-term plasticity alone, however in combination with a postsynaptic component, it helped restrain positive feedback, contributing to activity homeostasis. In conclusion, expression locus may determine affinity for distinct coding schemes while also contributing to keep activity within bounds. Our findings highlight the importance of carefully implementing expression of plasticity in biological modelling, since the locus of expression may affect functional outcomes in simulations.

## Introduction

Long-term synaptic plasticity is widely thought to underlie learning and memory as well as developmental circuit refinement [1]. The notion that synaptic plasticity underpins memory is typically attributed to Hebb [2], although for example Ramon y Cajal and William James had similar ideas long before Hebb [3].

After the discovery by Bliss and Lømo [4] of the electrophysiological counterpart of Hebb's postulate, now known as long-term potentiation (LTP), much effort has been focused on establishing the induction and expression mechanisms of long-term plasticity. In the 1990s, this led to a heated debate on the precise locus of expression of LTP, especially in the hippocampal CA1 region, with some arguing for postsynaptic and others for presynaptic expression [5]. Some early studies, however, favored a more nuanced view, e.g., by revealing that in hippocampal CA3 pyramidal cells, induction and expression of plasticity depended on synapse type [6]. Beginning in the early 2000's, the controversy was gradually resolved by the realisation that the details of plasticity depend on factors such as animal age, induction protocol, and brain region [7–9]. Currently, it is for example widely accepted that specific interneuron types have different forms of long-term plasticity [10, 11], which means long-term plasticity depends on synapse type, since synapses originating from the same axon may have distinct forms of plasticity depending on the target cell type [12]. Given the distinct functions of different synapse types, the diversity of expression mechanisms should perhaps not surprise [13]. Even so, the functional benefits of pre- versus postsynaptically expressed plasticity remain largely unknown, as they have only been explored in a handful of theoretical studies [14–19].

Going back several decades, a multitude of highly influential computer models of neocortical learning and development have been proposed, some of them focusing on aspects such as the dependence of induction on firing rates [20–22], while others have emphasised the role of the relative millisecond timing of spikes in connected cells [23–25], and some have included both [26]. Regardless of which factors determine plasticity in theoretical models, it has typically been the case that—with a few notable exceptions [16, 17, 19]—the expression of plasticity has been implemented as a simple synaptic weight change. As a minimal description, this is reasonable, since it is parsimonious to assume that long-term plasticity manifests itself as altered synaptic weights.

However, the expression of plasticity is not always well modelled by this sole change of synaptic weight. This is because presynaptically expressed plasticity leads to changes in synaptic dynamics, whereas postsynaptic expression does not (Fig 1). For instance, during high-frequency bursting, as the readily releasable pool of vesicles in a synaptic bouton runs out, leading to short-term depression of synaptic efficacy [27], while short-term facilitation may dominate at other synapse types [28]. Such short-term plasticity is important from a functional point of view because it acts as a filter of the information that is transmitted by a synapse [29–31]. Short-term depressing connections are more likely to elicit postsynaptic spikes due to brief non-sustained epochs of activity, whereas facilitating synapses require that presynaptic activity be sustained for some period of time to elicit postsynaptic spikes. In other words, short-term facilitating connections act as high-pass filtering burst detectors [32, 33], while short-term depression provides low-pass filtering inputs more suitable for correlation detection and automatic gain-control [34–36]. For example, increasing the probability of release by the induction of LTP would lead to more prominent short-term depression due to depletion of the readily-releasable pool, and as a consequence to a bias towards correlation detection at the expense of burst detection [37, 38].

Experimentally, it is long known that the induction of neocortical long-term plasticity may for example alter short-term depression [14, 39]. Although the functional consequences of short-term plasticity itself are quite well described [37, 40], the theoretical implications of *changes* in short-term plasticity due to the induction of long-term plasticity are not well explored. Yet, a majority of theoretical studies of long-term plasticity assumes that synaptic amplitude but not synaptic dynamics are altered by synaptic learning rules. One of the

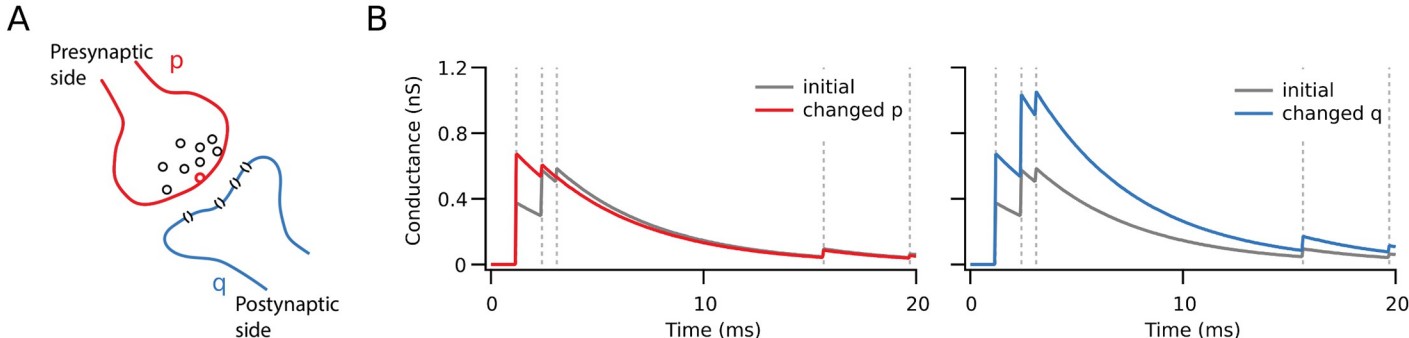

**Fig 1. The postsynaptic response to the same stimulus after plasticity depends on the locus of expression. (A)** Representation of pre- (red) and postsynaptic (blue) sides of a synapse, with probability of vesicle release *p*, and quantal amplitude *q*, i.e. the amplitude of postsynaptic response to a single vesicle. **(B)** Example of the difference between pre- and postsynaptic expression at inputs onto a cell. The identical initial response is illustrated in grey, while the potentiated responses are coloured red or blue. The amplitude of the first response after learning was set to be the same after pre- (red) and postsynaptic (blue) potentiation. With postsynaptic potentiation, the gain was increased by the same amount for all responses in the high-frequency burst. With presynaptic potentiation, however, the efficacy of the response train was redistributed toward its beginning, enhancing the first response but not the last.

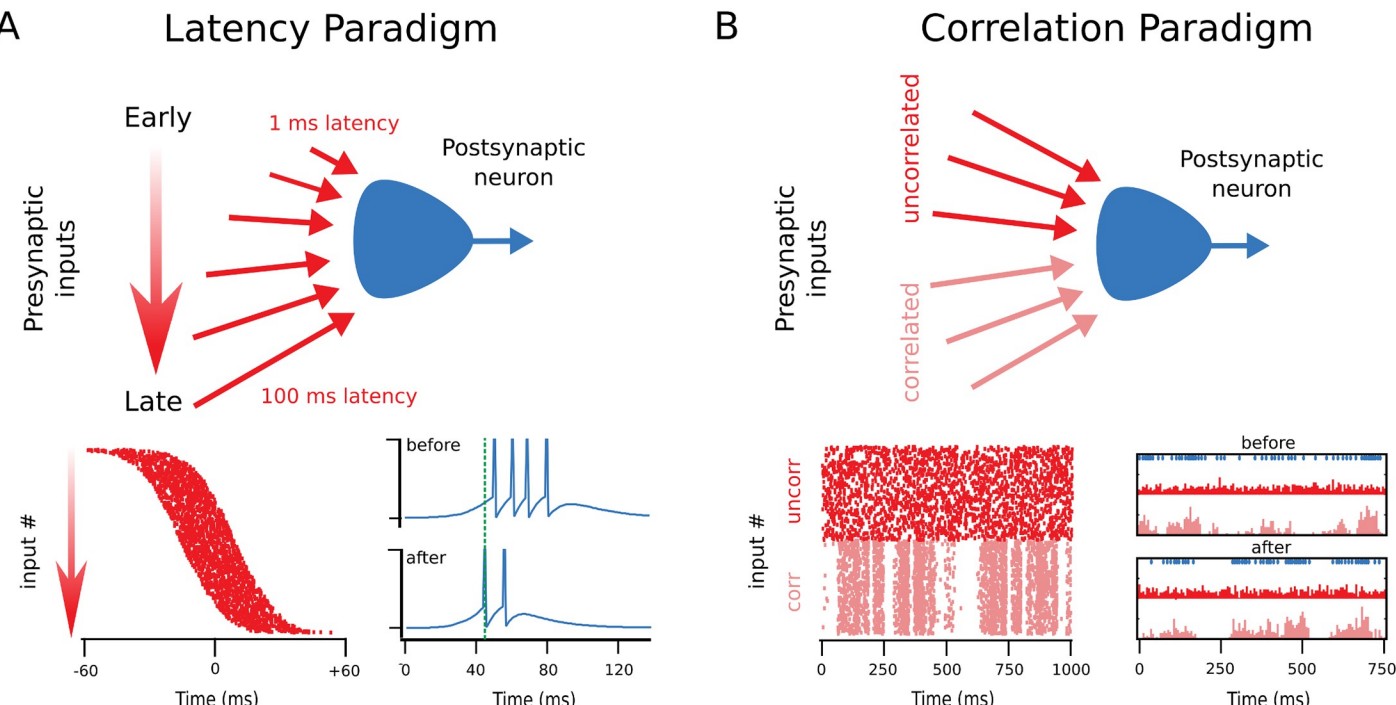

**Fig 2. Two different STDP learning paradigms were explored. (A)** Inputs arriving with a gradient of early to late timings resulted in reduced latency of the postsynaptic spiking response after STDP, as previously described [24]. In each trial, the postsynaptic neuron repeatedly received a brief volley of stimuli, between which short-term plasticity variables were allowed to return to their initial resting values. Bottom, left: Each presynaptic spike (raster dots) arrived with a different delay in the volley. Bottom, right: After a period of learning, the postsynaptic spiking response (blue) was shortened and started earlier, an expected outcome that was previously demonstrated [24]. **(B)** Correlated inputs were selectively potentiated by STDP, as previously described [43]: The postsynaptic neuron received persistent stimulation, with half of the inputs having correlated activity, while the rest were uncorrelated. Bottom, left: Raster plot illustrating the correlated (corr) and uncorrelated (uncorr) input spiking. Bottom, right: After learning, the postsynaptic spiking (blue raster at top) was more correlated with the correlated inputs (pink histograms) than it was before learning, reflecting how correlated inputs potentiated while uncorrelated inputs depressed. The outcome of this learning scenario is thus a selection for inputs that are correlated at the expense of those that are not [43]. In both paradigms, STDP was modelled with the same parameters (see Methods).

motivations of our present study is the observation that this seemingly innocuous assumption may not be neutral, but may in effect introduce a bias, because changing synaptic weight in theoretical models of long-term plasticity is equivalent to assuming that synaptic plasticity is solely postsynaptically expressed. This begs the question: What are the functional implications of pre- versus postsynaptically expressed long-term plasticity? Providing answers to this central issue is important for understanding brain functioning, as well as for knowing when weight-only changes in computer modelling are warranted.

Here, we use computational modelling to explore the consequences of expressing plasticity pre- or postsynaptically in a single neuron under two simple paradigms (Fig 2). One paradigm explores the postsynaptic response in relation to a repeated time-locked stimulus [24, 41, 42], while the other investigates the neuron's ability to detect a correlated stimulus [43–45]. Initially, we compare and contrast relatively artificial scenarios, for which the locus of expression is either solely presynaptic, solely postsynaptic, or equally divided between both sides. We then move on to investigating the functional impact in a biologically realistic model with separate pre- and postsynaptic components that were tuned to experimental data from connections between neocortical layer-5 pyramidal cells [17]. We report that presynaptically expressed plasticity adjusts the speed of learning, while postsynaptic expression is more efficient at

regulating spike timing and frequency. We conclude that pre- and postsynaptically expressed plasticity enable different complimentary functions and are not equivalent.

## Results

From a conceptual point of view, a synapse receives the output of a presynaptic neuron and transforms it into an input for the postsynaptic neuron. Most phenomenological models implement this by scaling the signal amplitude by a specific value, or 'weight'. However, it is known that a presynaptic action potential doesn't always elicit an output, which means this transmission is unreliable. The probability of transmission is largely determined by the probability of vesicle release from the presynaptic side. We take these two factors in a minimalist model of a synapse (Fig 1A). Thus the effective synaptic weight, $W$, is composed of a presynaptic part, $P$, and a postsynaptic part, $q$, so that $W = Pq$ (see Methods).

Postsynaptically expressed plasticity is readily implemented as a simple change in synaptic gain, by adjusting the quantal amplitude, $q$. The impact of postsynaptic expression is therefore relatively unambiguous, since it scales all postsynaptic responses the same way. For example, in the case of repeated measures of presynaptic stimulation, the standard deviation and the mean of synaptic responses scale the same, so the coefficient of variation remains the same [46], which means synaptic noise levels remain the same after postsynaptically expressed plasticity.

Presynaptic plasticity, however, has at least two different distinct types of impact on a synapse. First, the reliability and noise levels of neurotransmission are altered by presynaptic plasticity, because vesicle release is stochastic. Assuming release is binomially distributed, increasing the probability of release, $p$, typically increases the mean of synaptic responses considerably more than the standard deviation, which means that—for physiologically relevant initial values of $p$—the coefficient of variation is typically decreased by presynaptic LTP [46]. Second, increasing the probability of release depletes the readily releasable pool of vesicles more rapidly. Therefore, synaptic short-term dynamics are necessarily changed by presynaptically expressed long-term plasticity, resulting in functional differences.

To limit the scope of the study, we focus on early forms of plasticity for which we have detailed experimental data [17]. We thus do not consider the possibility that the number of release sites, $n$, may change, as it does in late, protein-synthesis dependent forms of plasticity [47].

We furthermore decompose presynaptic plasticity to distinguish between two distinct types of impact: unreliable transmission without short-term dynamics, or with short-term dynamics. We start with presynaptic expression modelled as direct changes in the probability of vesicle release (*without* changes in short-term plasticity) and compare that to postsynaptic expression. Subsequent to that, we model presynaptic expression as changes in short-term plasticity and compare to postsynaptic expression. This way, we aim to systematically tease apart two kinds of contributions of presynaptically expressed plasticity, i.e., changes in stochastic release versus changes in synaptic short-term dynamics.

### Presynaptic expression modelled as changes in stochastic release

Here, changes in the presynaptic weight $P$ were explored in terms of their impact on stochastic release at connections onto a single-compartment point neuron (see Methods). In other words, the effects of changes in $P$ on short-term dynamics are *not* reported here, as only the vesicle release probability $p$ was affected by LTP (we thus set $p = P$); we revisit that aspect in the next section.

With the latency paradigm, STDP leads early inputs to potentiate and late inputs to depress. In this paradigm, a volley of stimuli arrives at the postsynaptic neuron with varying delays (Fig 2A), plasticity therefore resulted in the shortening of the time to respond—the latency—of the postsynaptic neuron, as well as a temporal sharpening of the response, with fewer spikes and shorter inter-spike intervals [24]. The average latency reduction (Fig 3A and 3B), as well as the overall distribution of synaptic weights, decrease of postsynaptic activity duration and increase of postsynaptic firing frequency (Fig 3C, 3D and 3E) did not differ appreciably with the locus of plasticity. In comparison to the purely postsynaptic case, simulations with presynaptic plasticity presented a smaller variance of the latency shift across simulations (Fig 3B, inset). Potentiation also developed faster with presynaptic expression (Fig 3F). This can be framed as a consequence of potentiation requiring glutamate release [48], so that in a more reliable synapse, with a high $p$ value, there is a greater propensity for potentiation. Conversely, depression was slower with presynaptically expressed plasticity, again because lowered probability of release effectively also led to less plasticity (Fig 3F).

Next, we explored the correlation paradigm, in which plasticity selectively potentiates correlated inputs (Fig 2B) [43]. Here, all plasticity implementations detected the input correlations. However, presynaptically expressed plasticity generally promoted faster learning, e.g. synaptic weights evolved more rapidly (Fig 3G), similar to what we found above for the latency paradigm. However, there were exceptions to this general observation—for strong correlations, postsynaptic plasticity was faster to potentiate for correlated and faster to depress for uncorrelated inputs at certain input frequencies (Fig 3H). Which form of plasticity led to faster learning thus depended on the details of the firing statistics.

To explore this exception in more detail, we ran simulations where all of inputs were correlated, but half of them expressed plasticity only presynaptically, and the other half only postsynaptically. We imposed a limit to the total sum of weights so these two input populations competed, so that one potentiated at the expense of the other, which depressed. With this approach, we systematically explored the correlation-frequency space in distinct simulations where all inputs had a specified correlation and firing rate. We found that postsynaptic expression won for very highly correlated inputs for sufficiently low input frequencies (Fig 3I).

## Presynaptic expression modelled as changes in short-term plasticity

We next explored the effects of altering short-term dynamics (see Methods). This adds another aspect of presynaptically expressed plasticity, since short-term plasticity takes into account the history of presynaptic activity. In this scenario, presynaptic changes redistribute synaptic resources used over a limited time period, modulating vesicle release probability $p$ according to recent activity [14, 39]. Since $p$ is varying on a short timescale, the presynaptic weight in this case corresponds to the baseline value $P^B$ around which $p$ fluctuates (so $P^B = P$), so that $p = p(P^B, t)$. Even if the amplitude of an individual EPSP were affected equally by pre- and by postsynaptically expressed plasticity, the total input from a burst would still differ dramatically depending on the site of expression (Fig 1B).

In the simulations with the timed input configuration, results differed considerably depending on the specific locus of plasticity in the latency configuration. Postsynaptic expression alone provided the largest latency reduction, and also achieved it faster than the other plasticity implementations (Fig 4A and 4B). With presynaptic expression, the change in latency was smaller compared to the mixed setting with both pre- and postsynaptic expression, for which results may vary between extremes according to the ratio of pre- and postsynaptic expression. Effects of postsynaptic plasticity over response duration and intraburst frequency (Fig 4C and 4D) were also more marked, as expected from a higher integrated input (Fig 1B). The

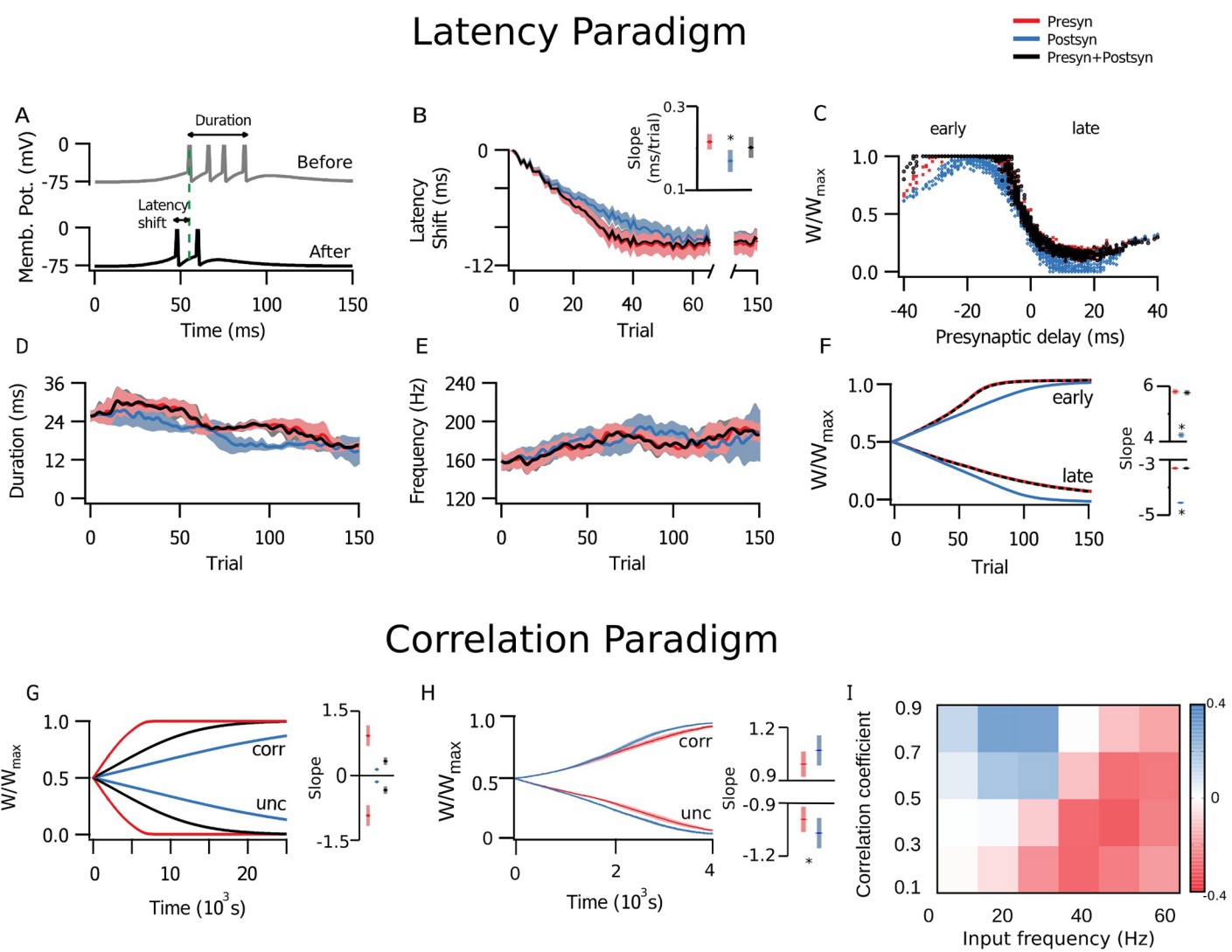

**Fig 3. With stochastic release, presynaptic plasticity typically promoted faster learning. (A-F)** Simulations in the latency paradigm (see Fig 2A). **(A)** Sample postsynaptic traces from trials before (grey) and after (black) plasticity. Initial response latency is marked by green dashed line. **(B)** STDP shortened the spike latency, as previously shown [24]. All graphs are colour-coded: only presynaptic plasticity (red), only postsynaptic plasticity (blue), or both pre- and postsynaptic plasticity (black) are implemented, with lines denoting the average of 10 independent realizations and the shading the standard error of the mean (SEM). Inset: presynaptic plasticity was faster than postsynaptic plasticity alone (t-test, p-value = 0.008). **(C)** Synaptic weight distribution after 150 trials, normalized and sorted relative to the fixed presynaptic delay. **(D, E)** Postsynaptic response duration (i.e, the interval between first and last spike in each trial) and the burst frequency did not differ for different expression loci. **(F)** Time evolution of average synaptic weight among early and late presynaptic inputs (i.e., input cells that spiked in the first or the second half of the stimulus) show how post-only expression (blue) was slower for the early group. Inset shows linear slope ($x10^{-3}$ /trial) across the first 100 trials. **(G-I)** Simulations in the correlation paradigm (see Fig 2B). **(G)** Potentiation and depression of the average synaptic weight among correlated inputs was faster in the presynaptic case. Inset shows linear slope ($x10^{-4}$ /s) across the first 50 seconds. **(H)** However, for highly correlated inputs (c>0.9), learning was faster with postsynaptic expression. This indicated that which form of plasticity led to faster learning depended on the details of the input firing pattern. Inset shows linear slope ($x10^{-4}$ /s) across the first 50 seconds. **(I)** The map shows the difference of $P$ and $q$ at the end of simulations. All inputs were correlated, but half expressed plasticity presynaptically, and the other postsynaptically. We found that across the explored parameter space, the half with presynaptic expression (red) typically won out, although the half with postsynaptic expression (blue) was victorious for a smaller parameter space where input firing frequency was low and correlations quite high.

simulations with both sides changing appeared closer to either the presynaptic case (duration, Fig 4C) or the postsynaptic case (frequency, Fig 4D). Here, changes in $P$ had a relatively greater influence on response duration, while changes in $q$ had greater impact on the response frequency. Nevertheless, synaptic efficacy was still potentiated faster and depressed slower in the presynaptic case (Fig 4E). This was similar to the above stochastic release implementation of

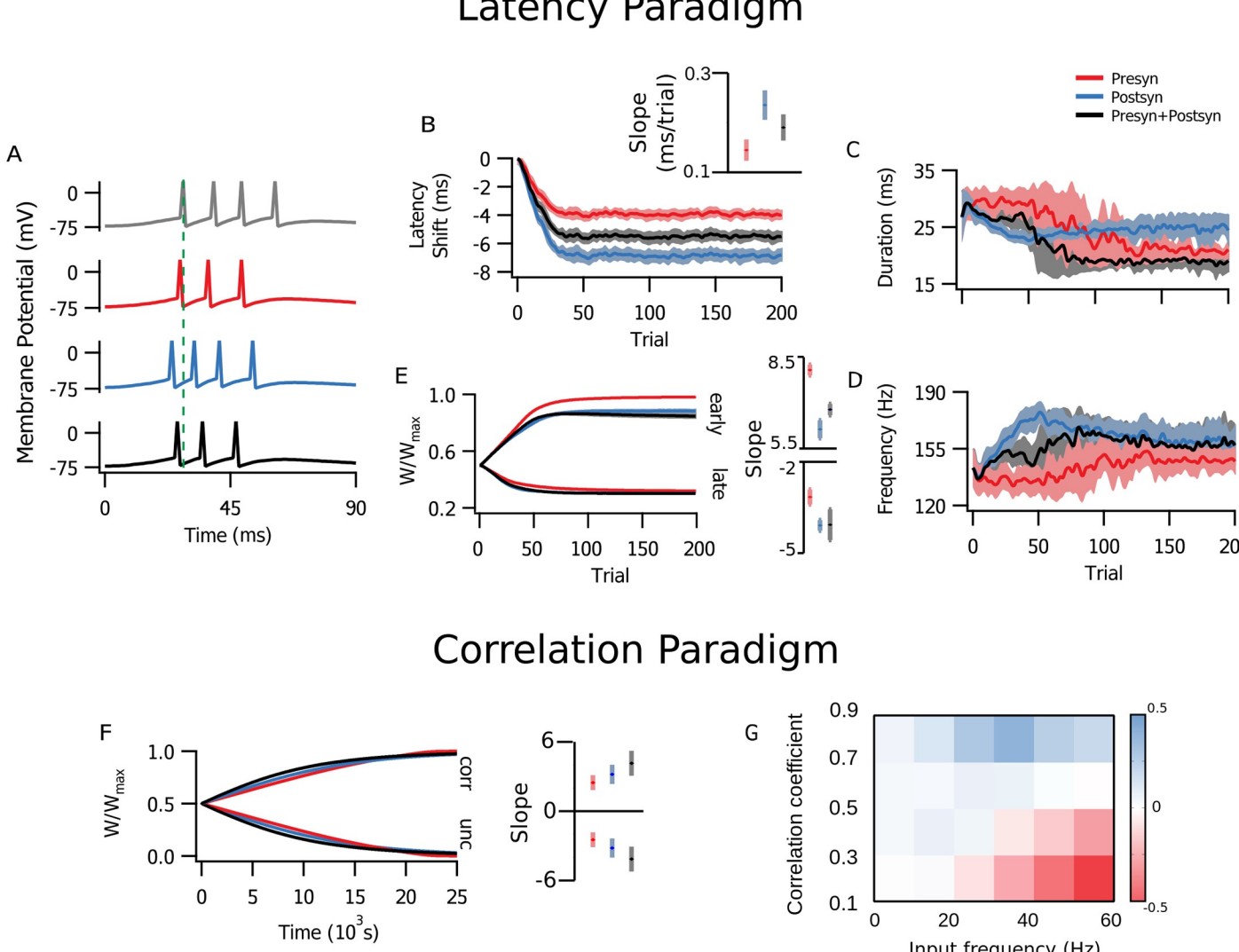

**Fig 4. Altering short-term plasticity was less efficient at reducing postsynaptic latency. (A-E)** Simulations in the latency paradigm (see Fig 2A) are colour-coded: red denotes presynaptic plasticity alone, blue postsynaptic plasticity alone, and black combined pre- and postsynaptic plasticity. Lines denote the average across 10 realizations, and the shading the SEM. **(A)** Example traces of postsynaptic activity before (grey) and after plasticity (coloured). Initial response latency is illustrated by the vertical dashed line. **(B)** Latency reduction was both faster and more marked for postsynaptic (blue) than for presynaptic (red) or combined (black) plasticity. Inset: The slope of latency reduction was steeper when postsynaptic expression was involved (t-tests: between pre- and postsynaptic expression, p-value $< 10^{-6}$; between presynaptic expression and both, p-value = 0.0008, between postsynaptic expression and both p-value = 0.003) **(C)** Combined and presynaptic plasticity reduced response duration more than with postsynaptic expression alone. **(D)** Burst frequency was similarly increased with all three forms of plasticity, although rate change was faster with postsynaptic plasticity. **(E)** Time course of average synaptic weights for early (left) and late (right) inputs. Inset shows linear slope ($x10^{-3}$ /trial) across the first 50 trials. **(F, G)** Simulations in the correlation paradigm (see Fig 2B) **(F)** Time course of average synaptic weights for correlated (left, "corr") and uncorrelated (right, "unc") inputs were largely indistinguishable across plasticity loci. Inset shows linear slope ($x10^{-5}$ /s) across the first 100 seconds. **(G)** As with Fig 3I, colour represents the difference between $P$ and $q$. This map of competition between input populations with pre- or postsynaptically expressed plasticity indicated a less marked differentiation except for very high (0.9) or very low (0.1) correlation coefficients.

presynaptically expressed plasticity, although it was less pronounced. This means that even if the rate of learning was effectively faster, presynaptic expression affected latency less rapidly than postsynaptic expression did (Fig 4F).

On the other hand, under modulation of short-term plasticity, plasticity rates in the correlated inputs paradigm evolved differently compared to the above stochastic release

implementation (Fig 4G). In the simulations where long-term plasticity affected short-term plasticity, the rate of change was slightly faster with postsynaptic than with presynaptic plasticity.

These findings show that the outcome in the latency paradigm was more affected by the locus of expression of plasticity than in the correlation paradigm. In conclusion, computational advantages could be tailored to optimally achieve a specific functional outcome by recruiting pre- or postsynaptic plasticity differentially.

## Comparisons with a biologically tuned model

The above minimalist toy models had the advantage that they provided full control of several key parameters. However, the relevance of the findings for the intact brain was unclear. To address this shortcoming, we explored the biological plausibility in a model [17] (see Methods) that was fitted to long-term synaptic plasticity data obtained from connections between rodent visual cortex layer-5 pyramidal neurons [39, 49, 50]. We could thus to some extent verify whether the results obtained with the minimal models hold in a more complex, data-driven context. We want to clarify upfront that in this model, LTP is expressed both pre- and postsynaptically, whereas LTD is solely presynaptically expressed. This asymmetry may seem odd, but it is derived from experimental data, and we have previously found that this arrangement provides certain computational advantages [17].

We first explored the latency paradigm (Fig 2A). To avoid disrupting the parameter tuning, instead of normalising the total synaptic change on each side, we kept the data-derived ratios and blocked either pre- or postsynaptic changes. Even so, we found that both pre- and postsynaptic plasticity components independently led to the shortening of postsynaptic latency (Fig 5A–5C). As with the earlier toy models that were not biologically tuned, postsynaptic changes appeared to affect spike latency more. Thus, looking at the case with both pre- and postsynaptic plasticity, postsynaptic potentiation essentially helped to reduce the latency compared to presynaptic plasticity alone, but pre- and postsynaptic plasticity together were slower than postsynaptic plasticity alone (Fig 5B).

In keeping with experimental results [39, 50]—which showed presynaptic LTP, presynaptic LTD, and postsynaptic LTP, but no postsynaptic LTD—the tuned model lacked the capacity for postsynaptically expressed depression. As a consequence, postsynaptically expressed potentiation led to inflated postsynaptic frequency and duration when implemented alone (Fig 5D and 5E). However, the presynaptic LTD was enough to produce a temporally sharpened response of shorter duration. With postsynaptic plasticity, the dynamics developed faster (Fig 5F), a result of a positive-feedback loop arising from increased postsynaptic firing rates (compare Fig 5A).

In the correlation paradigm (Fig 2A), groups of correlated and uncorrelated inputs clustered (Fig 6A) without the need for added competition through weight normalization [44, 51]. This only occurred when both pre- and postsynaptic plasticity components were implemented, and was not achieved through other models with physiologically compatible parameters [45].

To better understand the robustness of this property, we quantified the capacity of separation between correlated and uncorrelated populations with a linear separator. It was trained to classify inputs as correlated or uncorrelated according to the average and variance of $W$ values (Fig 6C). The presynaptic frequency range for optimal separation was between 50 and 80 Hz (Fig 6C). At the lower end of the range, it was bounded by the STDP correlation time scale of $\tau = 20$ ms (see Methods), meaning inter-spike intervals longer than 20 ms could not represent the minimal interval of correlation. At the upper end of the range, the high presynaptic

# Latency Paradigm

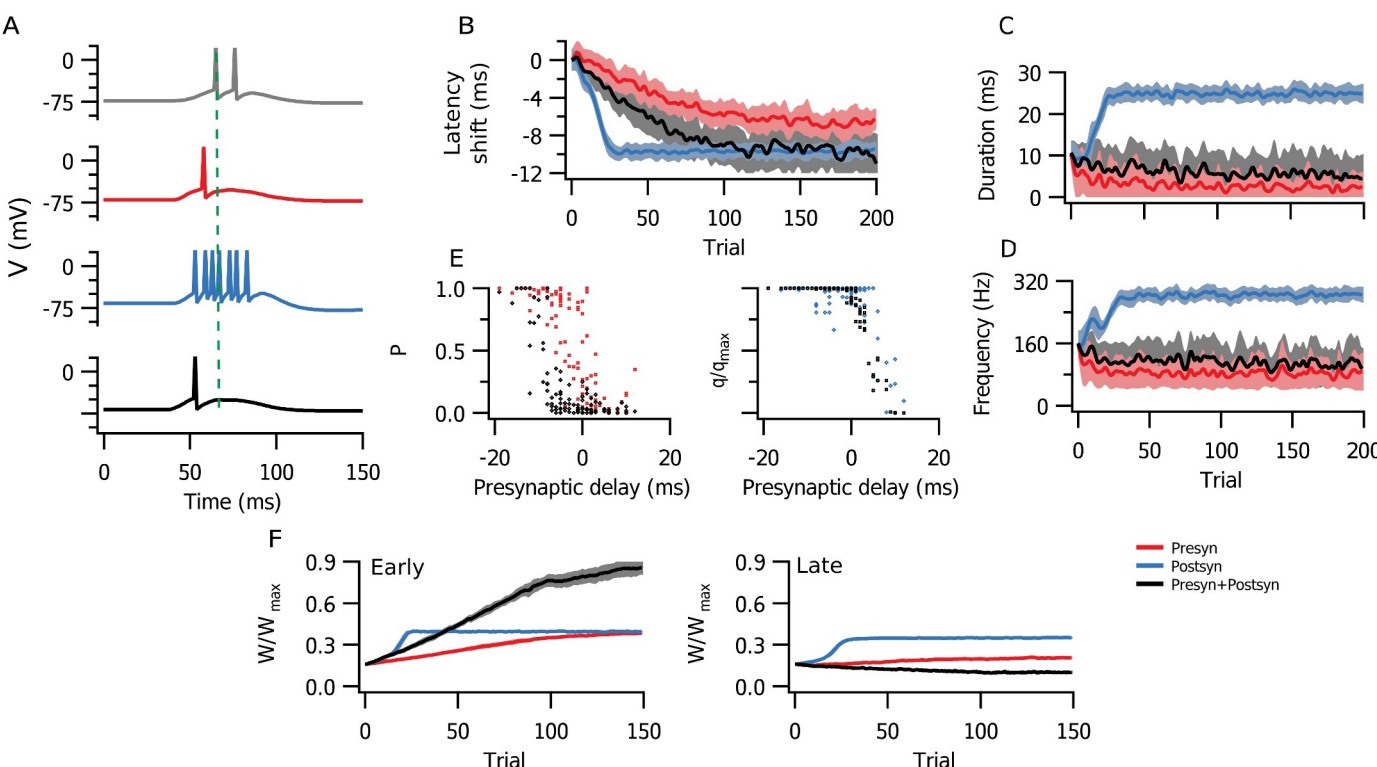

**Fig 5. A biologically tuned model verified key findings obtained with minimalist models.** (A) Sample traces of postsynaptic activity before (grey) and after only presynaptic (red), only postsynaptic (blue), or both pre- and postsynaptic learning (black). Lines indicate the average across 10 realizations, and the shading the SEM. The initial response latency is indicated by the green dashed line. (B) The postsynaptic response latency was shortened by learning, although both faster and more efficiently with postsynaptic learning. (C, D) Changes in duration and burst frequency of postsynaptic activity mirrored those obtained with the stochastic minimalist models (Fig 4C and 4D). (E) Distribution of pre- (*P*) and postsynaptic efficacies (*q*) after 200 learning trials. (F) Average synaptic weight of early (left) and late (right) presynaptic inputs evolved in distinct manners, however (compare e.g. Fig 4).

frequency yielded overall potentiation that included uncorrelated inputs, limiting the separation from the more potentiated correlated population (S1 Appendix).

In the same way as in the latency paradigm (Fig 5D), postsynaptic potentiation increased postsynaptic firing rate (Fig 6B). However, presynaptic plasticity alone produced no such effect. In combination with postsynaptic plasticity, presynaptic plasticity helped to lower postsynaptic firing frequency as *q* saturated (Fig 6B), thus keeping postsynaptic firing rates within narrower bounds.

## Discussion

In recent years, it has become clear that diversity in LTP expression is both ubiquitous and considerable, depending on factors such as animal age, induction protocol, and precise brain region [7–9, 13]. In this work, we explored possible functional properties of either pre- or postsynaptic locus of plasticity expression, and found that even in a single neuron scenario overall dynamics may be affected by it. This is an important feature to be considered, as many theoretical studies have focused on induction but not many in the expression of plasticity. Plasticity

# Correlation Paradigm

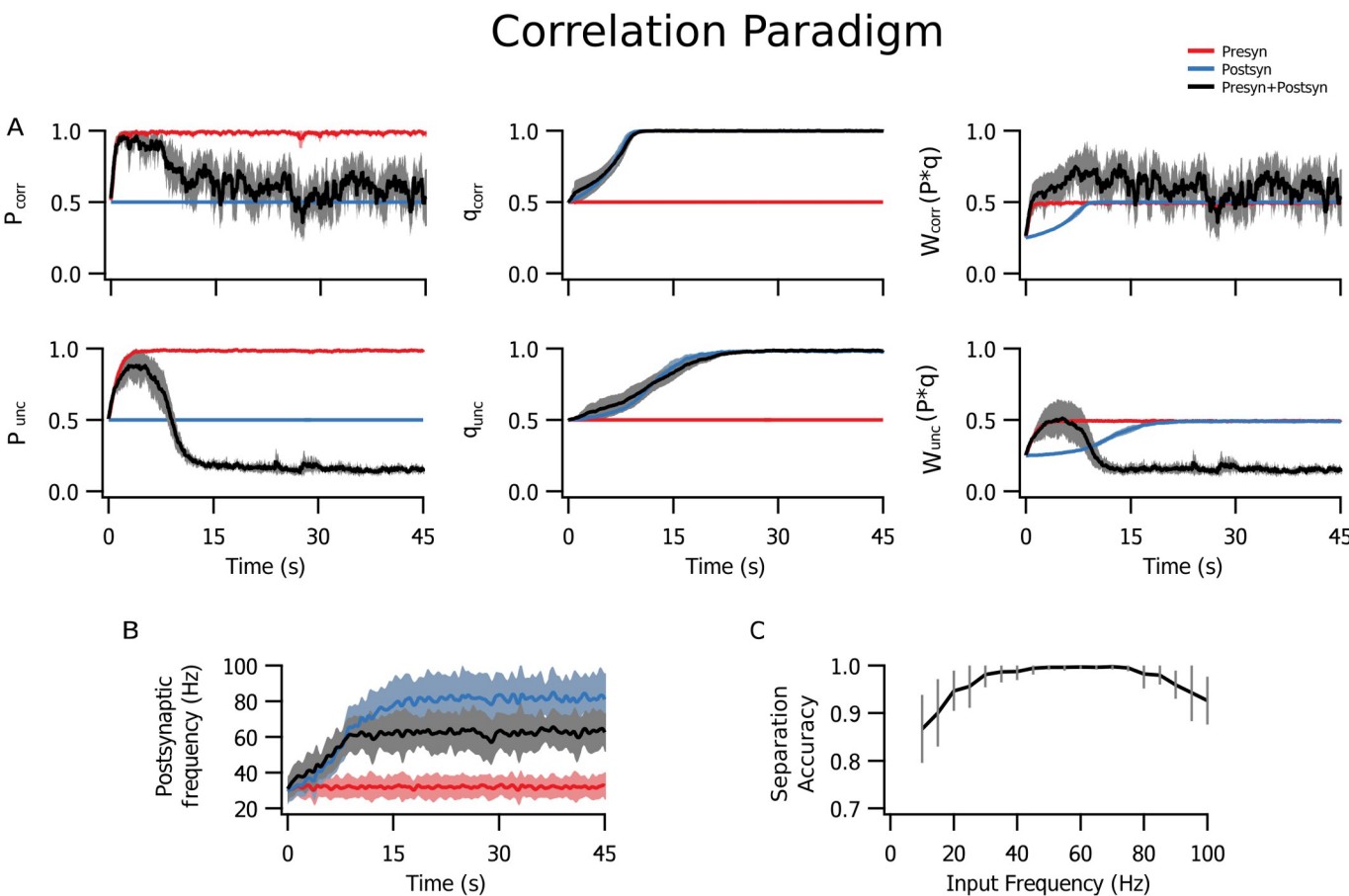

**Fig 6. The biologically tuned model clustered inputs with correlated and uncorrelated activity. (A)** Normalized averages (across 10 independent realizations) for presynaptic (*P*), postsynaptic (*q*) and combined pre- and postsynaptic (*W*) plasticity of correlated (corr) and uncorrelated (unc) inputs show that meaningful learning and segregation of inputs only occurred when both pre- and postsynaptic learning mechanisms were engaged. Surprisingly, presynaptic (red) or postsynaptic expression alone (blue) could not cluster differentially correlated inputs (*W*, right). **(B)** The postsynaptic spiking frequency increased when postsynaptic plasticity was engaged (blue and black), but not with presynaptic-only learning (red). **(C)** Average fraction of correct classifications between correlated and uncorrelated inputs for pre- and postsynaptic expression combined was optimal for presynaptic frequencies in the range 50 and 80 Hz.

has in the typical phenomenological model been implemented by default as a straightforward change in synaptic weight [24, 52, 53], although there are a few notable exceptions [14–16, 54, 55]. In other words, in the absence of better information, a standard assumption has been that the locus of expression does not matter appreciably for the modelling scenario at hand. Our findings challenge this standard assumption, highlighting how it may introduce a bias. For example, over-representation of postsynaptic expression may exaggerate the capacity to learn spike timing (e.g., Figs 4A and 5B).

We investigated two different learning paradigms, one with differently timed inputs, in which postsynaptic latency to spike was used as a measure of learning (Fig 2A), and another under constant stimulation, where a subset of inputs were correlated and potentiated together (Fig 2B). We first worked with simplified conceptual STDP models and later with a more realistic, biologically tuned model in which pre- and postsynaptic components were tuned to connections between neocortical layer-5 pyramidal cells [17].

## Pre- and postsynaptic expression favour different coding schemes

Our study showed that the locus of expression of plasticity determined affinity for different coding schemes. Presynaptic plasticity expressed as the regulation of release probability alone did not result in appreciable differences for steady-state postsynaptic activity compared to postsynaptic expression (Fig 3B). However, in the presence of short-term plasticity, presynaptic expression of long-term plasticity had a smaller impact on the spike latency than did postsynaptic expression (Fig 4B). This was because, as synaptic response amplitude grew, fewer inputs were needed to evoke a postsynaptic spike. With presynaptic expression, however, the spike still depended on the sum of a larger number of inputs. However, weight changes developed faster with presynaptic plasticity, thereby increasing the speed of learning. This effect, however, was not present in the correlation paradigm, where both pre- and postsynaptically expressed cases performed similarly.

Presynaptically expressed plasticity alone was not ideally suited for changing rate coding, because presynaptic short-term plasticity acts as a filter on the presynaptic firing rate. As a consequence, postsynaptic instantaneous firing frequency shows reduced changes (Figs 4D and 5D) or no changes (Fig 6B) when compared to postsynaptic plasticity. Presynaptic plasticity thus appeared to act as a limiter or a form of homeostasis for postsynaptic activity, in agreement with previously published interpretations [38]. The flip-side of this stabilizing feature of changes in short-term plasticity [56] is in other words the loss of ability to rate code well. An important cautionary take-home message from this observation is that the default implementation of plasticity as purely postsynaptic may thus lead to an erroneous overestimation of the impact on postsynaptic firing rates.

Frequently, the effect of unreliability of single synapses is considered to simply be one of noise or energy economy [57]. However, one can in fact consider this unreliability as a representation of uncertainty over a synaptic weight compared to its optimal value [58, 59]. It would then be plausible to consider presynaptic plasticity as an uncertainty tuning over the posterior distribution in a probabilistic inference framework [60].

## A biologically tuned model corroborated the toy model predictions

The same basic properties were observed in the biologically tuned model with simultaneous pre- and postsynaptic plasticity. Learning was dramatically affected by postsynaptic plasticity, while the presynaptic side appeared to act more on the rate of learning and on weight dynamics. It is possible that these results could be modified according to the ratio of pre- versus postsynaptic forms of plasticity, to optimally achieve a specific computational outcome. It is noteworthy that the biologically tuned model was also capable of separating groups of correlated and uncorrelated inputs without the need for a hard competitive mechanism.

## Experimental tests of model predictions

Since it is possible to specifically block pre- or postsynaptic STDP pharmacologically [39, 50], several of our findings related to the locus of expression of plasticity are possible to directly test experimentally. For example, at connections between neocortical layer-5 pyramidal cells, it is possible to block nitric oxide signalling to abolish pre- but not postsynaptic expression of LTP [50]. It is also possible to use GluN2B-specific blockers such as ifenprodil or Ro25–6581 to block presynaptic NMDA receptors necessary for presynaptically expressed LTD without affecting postsynaptic NMDA receptors that are needed for LTP [39, 61]. As a proxy for learning rate, one could explore *in vitro* how blockade of different forms of plasticity expression impacts the number of pairings required for plasticity, or alternatively how the magnitude of plasticity is affected for a given number of pairings [50, 53]. *In vivo*, the impact on cortical

receptive fields could similarly be explored. For example, we predict that receptive field discriminability is poorer when presynaptic LTP is abolished by nitric oxide signalling block-ade [17].

## Conclusions

Here, we have challenged the standard assumption that modelling synaptic plasticity as a weight change is neutral and unbiased. To do so, we relied on two classic STDP studies [24, 43], extending them with stochastic release and with short-term plasticity, and subsequently revisited our findings with a more physiologically realistic model [17]. We found that even in a simple feed-forward scenario, the locus of expression may have a surprising and considerable impact on learning outcome—e.g., the biologically tuned model could not properly segregate differentially correlated inputs if either pre- or postsynaptically expressed STDP was lost. We expect that these effects will only be greater in recurrent networks, where presynaptic plasticity at loops and re-entrant pathways will exacerbate the effects of changes in synaptic dynamics due to alterations of the accumulated difference. This additional level of complexity may in particular complicate very large recurrent network models [62, 63].

As our collective understanding of the expression of long-term plasticity has improved, it has become clear that the long-held notion that plasticity is expressed predominantly postsyn-aptically is erroneous [7–9]. Since presynaptic expression is still relatively poorly studied, our understanding of long-term presynaptic plasticity in health and disease needs to be generally improved [64]. Specifically, our study highlights the need for more detailed modelling of the role of the site of expression. It is clear that it has implications for information coding, be it spike based, rate based, or probabilistic. Therefore, in modelling long-term plasticity, choosing the location of changes in weight is a matter of gravity.

## Methods

### Neuron model

All of the simulations consisted of one postsynaptic neuron receiving a number of presynaptic Poisson inputs. In the first section, we used a simple leaky integrate-and-fire model defined by

$$\tau_V \frac{dV}{dt} = E_v - V(t) - g(t)(E_e - V(t)), \tag{1}$$

in which the membrane potential $V$ decayed exponentially with a time constant of $\tau_V = 20$ms to the resting value of $E_v = -74$ mV, and the threshold for an action potential was $V_{th} = -54$ mV. After each spike it was reset at $V_0 = -60$ mV with a refractory period of 1 ms.

Inputs were received with probability $p_j$ and increased the conductance-based excitatory contribution ($g$), with reversal potential $E_e = 0$ mV. An impulse with amplitude $q_j * q_{max}$ ($q \in (0, 1]$ and $q_{max}$ the maximal amplitude) was summed for each $l^{th}$ input received at time $t_j^l$ from the presynaptic neuron $j$, and decayed exponentially with a time constant of $\tau_g = 5$ms:

$$g(t) = \sum_{j,l} q_j q_{max} \Theta(t - t_j^l) e^{\left(\frac{t - t_j^l}{\tau_g}\right)}, \tag{2}$$

where $\Theta(x)$ is the Heaviside function. In the the last section, we used the adaptive exponential integrate-and-fire model [65] to reduce unrealistic bursting and to comply with the biological

tuning [17]:

$$\frac{dV}{dt} = \frac{1}{C}\left[g_L(E_L - V) + g_L\Delta_T e^{\left(\frac{V - V_T}{\Delta_T}\right)} - g_e V - z\right] \quad , \tag{3}$$

$$\tau_W \frac{dz}{dt} = c_z(V - E_L) - z \tag{4}$$

The corresponding parameters for a pyramidal neuron were $C$ = 281 pF, $g_L$ = 30 nS, $E_L$ = −70.6 mV, $\Delta_T$ = 2mV, $c_z$ = 4nS, $\tau_W$ = 144ms. Spiking threshold was $V_T$ = −50.4 mV, and after each spike $V$ was reset to the resting potential $E_L$ while $z$ increased by the quantity $b$ = 0.0805 nA (as in [65]).

## Stimulation paradigms

The postsynaptic neuron was in one of two stimulus paradigms. The first one was based on [24] and is referred to as the Latency Paradigm (Fig 2A). In every 375-ms-long trial, the postsynaptic cell received a volley of Poisson inputs that arrived with a specific delay, normally distributed around a time reference, for each specific presynaptic neuron. Each input lasted for 25 ms with a spiking frequency of 100 Hz. We measured the time to spike of the first postsynaptic spike in response to a bout of stimuli using the mean of the presynaptic delay distribution as a reference point. For clarity, in the Results, curves that represent latency shift, intra-burst frequency or burst duration were smoothed using a moving average filter with a window of three points.

The second paradigm was based on [43] and is referred to as the Correlation Paradigm (Fig 2B). This configuration consisted of continuous Poisson inputs with fixed frequency. However, half of the inputs had correlated fluctuations of activity, with a time window of $\tau_{corr}$ = 20 ms, while the other half was uncorrelated. Correlations were implemented using method described in [66]. An additional scenario with competition between pre- and postsynaptic plasticities, all inputs are correlated but half changes presynaptically and the other half postsynaptically. The total sum of weights was kept fixed so that competition was observable in a wide range of parameters.

## Additive STDP model

For the majority of the simulations we opted to implement STDP with the simple additive model proposed by Song and Abbott [24]:

$$\Delta W_{ij} = \sum_k \sum_l F(t_i^k - t_j^l)$$

$$F(x) = \begin{cases} c_{pot} \exp(x/\tau_{STDP}), & x < 0 \\ c_{dep} \exp(-x/\tau_{STDP}), & x \geq 0 \end{cases} \tag{5}$$

Each increment to the synaptic weights $W_{ij}$ (since there was only one postsynaptic cell, we consider $W_j = W_{ij}$ throughout this paper) was computed after a pair of pre- and postsynaptic spikes, $t_i$ and $t_j$, and the parameters were set to $\tau_{STDP}$ = 20ms, $c_{pot}$ = 0.005, and $c_{dep}$ = −0.00525. We separated the synaptic weight $W_j$ as a product between pre- and postsynaptic counterparts, baseline probability of release $P_j$ = (0, 1] and quantal amplitude $q_j$ = (0, 1] respectively, so that

$W_j = q_j P_j$. The probability of release was simulated in two different ways, one equivalent to regulating the probability of stochastic interactions and the other via short-term plasticity.

To enable comparison of convergence rates for different types of plasticity expression, we ensured that the weight change $\Delta W = W^{t+1} - W^t$ at time step $t$ was the same regardless of whether plasticity was expressed presynaptically, postsynaptically, or both. To achieve this, we normalised the weight changes so that if only $q$ was changed:

$$\Delta W^q = P^t(q^{t+1} - q^t) = P^t \Delta q^q \tag{6}$$

and similarly, if only $P$ was changed:

$$\Delta W^P = q^t \Delta P^P \tag{7}$$

The initial value of all simulations was the same for $P$ and $q$, so that in these cases $\Delta W^P = \Delta W^q \equiv d$. When expression was both pre- and postsynaptic, the amount $d$ was divided equally across $P$ and $q$ (so $\Delta P^{Pq} = \Delta q^{Pq} \equiv \Delta$) as follows:

$$\begin{aligned}
\Delta W^{Pq} &= P^{t+1} q^{t+1} - P^t q^t \\
&= (P^t + \Delta)(q^t + \Delta) - P^t q^t \quad .
\end{aligned} \tag{8}$$

Solving for $\Delta$ so that $\Delta W^{Pq} = d$:

$$\Delta^{Pq} = -\frac{1}{2}\left[(P^t + q^t) - \sqrt{(P^t + q^t)^2 + 4P^t d}\right] \tag{9}$$

We also kept the same range of total $W$ change as equal throughout the simulations. Since both start at the same initial value ($P^0 = q^0$), the largest possible change for $P$ or $q$ separately was $\Delta_{tot} = P^0(1 - q^0) = q^0(1 - P^0)$. For changing $P$ and $q$ simultaneously, we limited the maximal values $P$ and $q$ so that $\Delta W_{tot} = P^{TOP} q^{TOP} - P^0 q^0$ is also the same. In this case, $q^{TOP} = P^{TOP} = \sqrt{q^0} = \sqrt{P^0}$.

## Biologically tuned STDP model

We compared the results of the straightforward additive model to a slightly more complex STDP model that acts separately over pre- and postsynaptic factors [17]. Parameters were fitted to experimental data from connections between pyramidal cells from layer 5 of V1 [39, 49, 50]. The equations for pre- and postsynaptic changes followed:

$$\Delta q_j = c_+ x_{j+}(t) y_-(t - \epsilon) Y(t) \quad , \tag{10}$$

$$\Delta P_j = -d_- y_-(t) y_+(t) X_j(t) + d_+ x_{j+}(t - \epsilon) y_+(t) X_j(t) \quad . \tag{11}$$

where $X_j(t) = \sum_l \delta(t - t_j^l)$ is increased at each spike from the presynaptic neuron $j$ and $Y(t) = \sum_k \delta(t - t_i^k)$ at each spike from the postsynaptic neuron $i$. $\epsilon$ is to emphasise that $\Delta W$ was calculated before $x_{j+}$ and $y_-$ were updated, upon the arrival of a new spike. $y_+$ and $y_-$ are postsynaptic traces,

$$\frac{dy_+}{dt} = -\frac{y_+}{\tau_{y_+}} + Y \quad , \tag{12}$$

$$\frac{dy_-}{dt} = -\frac{y_-}{\tau_{y_-}} + Y \quad , \tag{13}$$

with decay times $\tau_{y_+}$ and $\tau_{y_-}$ respectively, and $x_{j+}$ was a presynaptic trace with decay time $\tau_{x_+}$:

$$\frac{dx_{j+}}{dt} = -\frac{x_{j+}}{\tau_{x_+}} + X_j \quad . \tag{14}$$

The parameter values were taken from [17]: $d_- = 0.1771$, $\tau_{y_-} = 32.7$ms, $d_+ = 0.15480$, $c_+ = 0.0618$, $\tau_{y_+} = 230.2$ms and $\tau_{x_+} = 66.6$ms. To avoid manipulation of the fitting, weight changes were not normalised in this case. To avoid the postsynaptic side being forever potentiated, a small scaling was introduced postsynaptically in each step: $\Delta q_j^{scaled} = \Delta q_j - \alpha < \Delta q >$, being $< \Delta q >$ the average change over all postsynaptic side and $\alpha$ a scale factor (0.5).

In the last section, we used a linear least-squares classifier to infer whether presynaptic inputs were correlated or uncorrelated. A linear model was fitted to separate the values of synaptic weight averages and variances from half of the inputs (labelled correlated or uncorrelated), and then used to classify the other half of inputs.

## Presynaptic factor

Presynaptic control of the probability of release per stimulus was implemented either as a Markovian process or as short-term plasticity, with presynaptic weight $P_j$. In the former case, probability ($p_j$) of stochastic neurotransmitter vesicle release followed a binomial distribution, and $p_j = P_j$. Based on the findings reported by [67], each presynaptic neuron had $N = 5$ release sites that functioned independently. In the second case, we considered a dynamic modulation of the EPSPs through short-term plasticity. The probability of transmission was decomposed into the instantaneous probability of release $p_j^v(t)$ and availability of local resources $r_j(t)$, so that $p_j = p_j^v r_j$. These two factors modulate transmission in a short term scale around a baseline value of release probability $P_j^B$, which makes $P_j^B = P_j$ in this STP scenario. The dynamics of $p_j^v(t)$ and $r_j(t)$ followed the model proposed by Tsodyks and Markram [68]:

$$\frac{dr_j(t)}{dt} = \frac{1 - r_j(t)}{\tau_D} - p_j^v(t)r_j(t)X_j(t) \quad , \tag{15}$$

$$\frac{dp_j^v(t)}{dt} = \frac{P_j^B - p_j^v(t)}{\tau_F} + P_j^B[1 - p_j^v(t)]X_j(t) \quad . \tag{16}$$

Depression and facilitation time constants, $\tau_D = 200$ ms and $\tau_F = 50$ ms respectively, were chosen as representative values for connections between pyramidal neurons [69]. The resulting short-term plasticity is mostly depressing, that is the resulting $p_j$ is lower than $P_j^B$ except for very low values of $P_j^B \lessapprox 0.3$ and high input frequencies [70].

## Supporting information

**S1 Appendix. Rate Model.**
(PDF)

## Acknowledgments

We thank Alanna Watt and Mark van Rossum for suggestions, help, and useful discussions.

## Author Contributions

**Conceptualization:** Beatriz Eymi Pimentel Mizusaki, Rui Ponte Costa, Per Jesper Sjöström.

**Data curation:** Beatriz Eymi Pimentel Mizusaki.

**Formal analysis:** Beatriz Eymi Pimentel Mizusaki, Sally Si Ying Li.

**Funding acquisition:** Beatriz Eymi Pimentel Mizusaki, Sally Si Ying Li, Rui Ponte Costa, Per Jesper Sjöström.

**Investigation:** Beatriz Eymi Pimentel Mizusaki, Sally Si Ying Li, Per Jesper Sjöström.

**Methodology:** Beatriz Eymi Pimentel Mizusaki, Rui Ponte Costa, Per Jesper Sjöström.

**Project administration:** Per Jesper Sjöström.

**Resources:** Rui Ponte Costa, Per Jesper Sjöström.

**Software:** Beatriz Eymi Pimentel Mizusaki, Sally Si Ying Li, Rui Ponte Costa.

**Supervision:** Per Jesper Sjöström.

**Validation:** Beatriz Eymi Pimentel Mizusaki.

**Visualization:** Beatriz Eymi Pimentel Mizusaki, Sally Si Ying Li, Per Jesper Sjöström.

**Writing – original draft:** Beatriz Eymi Pimentel Mizusaki, Rui Ponte Costa, Per Jesper Sjöström.

**Writing – review & editing:** Beatriz Eymi Pimentel Mizusaki, Rui Ponte Costa, Per Jesper Sjöström.

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
