## [Decision Letter · Decision Letter 0]

21 Oct 2021

Dear Dr Sjöström,

Thank you very much for submitting your manuscript "Pre- and postsynaptically expressed spike-timing-dependent plasticity contribute differentially to neuronal learning" for consideration at PLOS Computational Biology.

As with all papers reviewed by the journal, your manuscript was reviewed by members of the editorial board and by several independent reviewers. In light of the reviews (below this email), we would like to invite the resubmission of a significantly-revised version that takes into account the reviewers' comments.

We cannot make any decision about publication until we have seen the revised manuscript and your response to the reviewers' comments. Your revised manuscript is also likely to be sent to reviewers for further evaluation.

Sincerely,

Hugues Berry

Associate Editor

PLOS Computational Biology

Kim Blackwell

Deputy Editor

PLOS Computational Biology

Reviewer's Responses to Questions

**Comments to the Authors:**

Reviewer #1: The manuscript addresses a relevant and interesting questions, aiming to distinguish the effects of changes in synaptic strength via increase/decrease in presynaptic vesicle release probability versus via an increase/decrease in postsynaptic receptor density. The former process impacts the short-term dynamics of vesicle release, such that long-term changes in synaptic strength are likely to have concomitant changes in the ratio of facilitation/depression. Since these changes that are overlooked in most/all models of plasticity in networks, the question is important.

Overall, the strategy used is clear and good, as the authors address a couple of roles of plasticity in a simple setting and compare models with and without the different contributions to changes in synaptic strength. I do find some of the specifics to be unclear though, in particular, I found insufficient description of the precise model when the authors attempt to isolate the facilitation/depression dynamics from the presynaptic strength-change – that is, how Fig 4 is achieved in comparison to Fig 3.

Fig 2 Caption and throughout: “The learning task” is a misnomer as there is no animal performing a task who can learn anything. Rather the paper is assessing how patterns of inputs give rise to different outputs in a circuit. There is no goal or “right answer”. Moreover, connections do not learn they change even if those changes almost certainly underlying an animal’s learning. Perhaps the paper could be reframed to particular tasks that animals must learn and the simple circuits be proposed to underly such learning. I don’t think that is necessary and just an alteration in the language used would be fine.

l. 156 “We next explored the effects of altering short-term dynamics”. The statement is confusing as the previous section already explored these effects via changes in “p”. Do you mean “altering short-term dynamics *without* any concurrent change in synaptic strength? Or do you mean while the change in synaptic strength is matched to the postsynaptic plasticity? It is really unclear to me, even when looking over the methods. Moreover, however this is done, how doe “postsynaptic alone” in this section (and Fig 4) differ from that in the prior section (and Fig 3) since “postsynaptic alone” could not impact short-term dynamics in either section. The “combined pre and post” becomes even more complicated to think about as distinct from the prior section or from “post alone” if short-term dynamics are present/missing. All rather confusing, so please clarify in the main text as well as the methods!

l. 198-214. There seems to be some confusion in the text here. If “LTD is solely postsynaptically expressed” then why in l.211 does “postsynaptic plasticity lack the ability to depress”?? And those problems resolved by inclusion of “presynaptic LTD” in the biological model if LTD is not presynaptic? I think some switching of “pre” and “post” in the text is needed and ne sure it matches Fig 5.

Fig 6. It is confusing that for the red and the blue traces in A there is essentially no difference between the “unc” and the “corr” results so there is no clear reason as to how inputs are segregated?

And how is this compatible with the classic papers of Song & Abbott showing separation of strengths for “corr” and “uncorr” inputs with postsynaptic plasticity alone? (Blue lines in your figure converge to identical values).

And if weights are kept constant in the red/blue traces then why are they not kept constant in the black trace (far right)? All rather confusing.

l. 317 “considerable impact on the learning outcome”: Maybe I missed this, but I saw no overall difference, just a change in the speed at which changes arose in the correlation paradigm – though a change in latency was a difference, but I think “small impact” with no qualitative change and it is hard to see what “learning outcome” for an animal is particularly impacted here if you wish to match these results to learning.

l.343-4 Eq.2: Why is the conductance only impacted by “q” rather than release probability “p”? Especially since in the rate model the connection strength “W” is proportional to both “p” and “q”.

l. 367-8 Eq.5: Why is the function “F” a constant rather than exponentially decaying? As written the rate of change of W depends on all prior (and future) spikes throughout all of time as they each contribute the same constant amount. This seems unphysical.

l. 376-7 “When the rate convergence rates were compared” It is unclear how you are comparing them when you are forcing them to be identical as you say you “ensure” they were “the same for all simulations”?

Minor:

Fig 3 caption: two “and”s near the end.

l.136 just “2B” I think

l. 185 “I”

l. 192-3 “relevance … was unclear”

l. 318 “these effects”

l. 377 I assume that the superscripts “f” and “i” denote “final” and “initial” here and elsewhere, but I don’t see where these are defined.

Fig 7. I suggest you make the arrows smaller so that they are separated – it is hard to tell which way they point when they run into each other, and the change in direction of the flow fields is important.

Reviewer #2: Here the authors present simulation results illustrating how the effects of spike-timing-dependent plasticity can vary depending on whether it is expressed pre- or postsynaptically. They focus on two scenarios: 1) a latency paradigm, where early inputs potentiate and late inputs depress to reduce the latency of responses of a postsynaptic neuron, and 2) a correlation paradigm, where a set of correlated inputs potentiates and sets of uncorrelated inputs depress causing the postsynaptic neuron to covary with the correlated inputs. In simulations of stochastic release, short-term plasticity, and a biologically tuned model they find that the locus of expression can alter the speed of potentiation/depression and several other outcomes for each paradigm.

This is a well-structured paper that makes a strong argument for the importance of thinking about locus of expression with plasticity. It develops interesting new ideas, and it will certainly be interesting to many computational and experimental neuroscientists.

I have no major concerns. The logic and arguments are clear. However, I did find the results section somewhat difficult to understand. Some details seemed missing or could otherwise be more clearly described here (see multiple points below).

Minor Issues:

Line 95: “First, in the reliability of transmission due to stochastic vesicle release.” Incomplete sentence?

Line 96: “keeping the standard deviation roughly the same” This seems unnecessarily vague, since you can write down the change in CV analytically. Why not just say that the standard deviation decreases (when initial p>0.5) or increases by a smaller amount than the mean

Line 110: “we model presynaptic expression as changes in short-term plasticity and compare that postsynaptic expression” Missing “to” unclear what “that” refers to.

Line 119: “Here, changes in pj were explored in terms of their impact on stochastic release” … the previous sentences make it seem like p and q will both be varied. I would also suggest spelling out what j is

Line 120: It may be helpful to have add some context here. E.g. “With the latency paradigm, STDP leads early inputs to potentiate and late inputs to depress.”

Line 129: What is “variance of the latency shift” referring to? Variance across simulations at a specific time point? The variance is difficult to see in the figure alone.

Fig 3 caption: “Time evolution of average synaptic weight among early and late presynaptic inputs (i.e., input cells that spiked in the first or the second half of the stimulus) show how post-only expression (blue) is relatively slower.” … This seems incomplete – isn’t postsynaptic expression slower for potentiation, faster for depression?

Fig 3: Suggest maybe using dashed lines to make it clear that pre follows pre+post in panel F

Fig 3H: It would be helpful to add “corr” and “unc” labels here, as well as an indicator that c>0.9 here. I would also suggest adding the value of c for 3G for context.

Fig 3I: a colorbar would be helpful here. Is this the difference in slopes? Weights? The caption seems to suggest it’s related to “which side potentiated faster”, but the text seems to focus on who “won”

Fig 3I caption: typo “and and”

Line 140: “was faster for strongly correlated input firing at certain input frequencies”. Should clarify that it’s faster for both correlated (faster to potentiate) and uncorrelated (faster to depress) inputs? Additionally, it’s not clear what the “input frequency” is here or that it matters.

Line 150: “we systematically explored the correlation-frequency space”. It’s somewhat unclear what this means. It may be useful to spell out that (if I understand correctly?) these are distinct simulations where all inputs have a specified correlation and firing rate.

Line 152: “a scenario that corresponds to fewer inputs spiking synchronously” I’m not sure what exactly this is comparing.

Line 167: “more subtle” I wasn’t completely sure what this meant. Just that the change in latency was smaller?

Fig 4 caption: “between p and q” and “between p and both”. Suggest spelling out pre or postsynaptic expression here to avoid confusion.

Line 182: “STP modulation” I guess this if the first use of the acronym. Could spell it out for clarity here.

Line 185: typo “I”

Line 187: “It thus appears that computational advantages could be tailored to a functional task at hand by recruiting pre- or postsynaptic plasticity differentially.” I’m somewhat confused by this conclusion. Unlike the results from Fig 3G, it seems like there’s minimal difference between the speeds when the dynamics are included. I’m unsure if/how 4G and 3H can be compared directly, but at low firing rates it also appears that the site of expression doesn’t matter.

Line 206: “As with the above simplistic modelling scenarios” I guess this is just the case with dynamics.

Line 207: “spike timing” Might specify that it’s “latency”, since other aspects of timing aren’t directly analyzed?

“when both pre- and postsynaptic plasticity were active, the presence of postsynaptic potentiation further reduced the latency compared to presynaptic plasticity alone.” This seems like an odd framing for modelers who traditionally only use postsynaptic plasticity. It may be useful to reframe by focusing on the fact that pre+post expression is much slower than post alone.

Line 225: It’s not clear to my why one would classify based on p rather than w.

Line 283: “It is also interesting to think about the role of presynaptic plasticity if it is not very useful in the context of usual ’coding’ frameworks.” To the extent that STP acts as a filter or mechanism for gain control wouldn’t presynaptic expression of long-term plasticity just reflect changes in the filter/gain control? This seems like a direct impact on coding.

Line 352: typo “Peduction”

Line 364: typo “implemente”

I couldn’t find how many inputs were used here or how many simulations are being averaged over in the results. I would also suggest explicitly describing the error bands/bars in the captions of figures where they appear (S.D., C.I. or S.E.M?).

Reviewer #3: In their paper Mizusaki et al address an important and timely question of how the site of expression of synaptic plasticity affects the outcome of learning. While we do know since work of Tsodyks & Markram that pre- and postsynaptic changes modify cell's responses to irregular inputs in completely different ways, detail of these differences and their computational consequences remain poorly explored. Systematic analysis of such differences would make an important contribution toward understanding specifics of learning in diverse neuronal networks.

However, while the topic of the study is interesting, and the paper could potentially make an important contribution to the field, there are several concerns that should be addressed.

In my opinion, major problems that needed to be addressed are:

1. Description of results (actually methods too) is neither precise nor complete; there are numerous unsubstantiated or unclear statements (specified below). Also, in the present version it is not clear whether and how model settings were changed in different experiments (see below); this should be clearly described.

2. Modeling presynaptic changes that do not lead to changes of synaptic dynamics looks odd. The rationale for using such a model, which contradicts almost everything we know about synaptic transmission, should be clearly articulated.

3. Results of simulations with Latency paradigm and Correlation paradigm should be presented to allow comparison. e.g. there is a lot of description of spiking response in latency paradigm, but not in correlation paradigm experiments.

Specific comments:

l.55 'depletion' – double

l.96-98 "Assuming release is binomially distributed, increasing the probability of release, p, 96

increases the mean of synaptic responses while keeping the standard deviation roughly 97

the same, ..." – this is plainly wrong.

With N=5 (Methods, l.408) and q=1; with p= 0.1, 0.2, 0.3 the mean is 0.083, 0.167, 0.25 and SD is 0.133, 0.197, 0.248. How 0.133, 0.197, 0.248 are "roughly the same"?

l. 108-112 "We start with presynaptic expression modelled as direct 108

changes in the probability of vesicle release and compare that to postsynaptic 109

expression. Subsequent to that, we model presynaptic expression as changes in 110

short-term plasticity and compare that postsynaptic expression."

Not clear what's the difference. If 'direct changes of p' means no short-term plasticity, it's a mathematical abstraction without any biological sense. At least I am not aware of a cortical synapse that does not express STP that depends on p.

Please either provide a clear rationale for using such a construct, or remove.

Fig 2 legend.

"(A) ... The learning task is thus to reduce the

latency and to shorten the duration of the postsynaptic spiking response"

" (B) ... The learning task

is thus to select for inputs that are correlated at the expense of those that are not"

'Learning task' sounds weird here. Whose task?

What is actually described, is how inputs with two fundamentally different types of temporal structure are affected by STDP.

"After learning, the postsynaptic spiking (blue raster at top) was more correlated

with the correlated inputs (pink histograms) than with the uncorrelated inputs (red

histograms), indicating that the former drove postsynaptic activity." Same was true before learning... It's kind of commonplace that more correlated inputs have stronger impact on postsynaptic firing, with or without plasticity.

Also, from this figure and its description it's not clear how two paradigms will be compared; How to compare a decrease of spike number and latency on the one hand, with a change of the correlation between inputs and spiking on the other?

l. 126. " decrease of postsynaptic activity duration and increase of postsynaptic firing frequency" – do not fit together.

Fig. 3

3B: X-scale labels are missing. Why is it different from A, D-F?

3G: what was correlation (strongly, weakly correlated inputs)?

3B,D-F X-scale is in trials; in G,H – in s; is it possible to make them uniform? To somehow compare the rate of synaptic changes?

3I – not sure I understand what and how was going on here. All inputs were saturated but some faster, some slower? Please show examples of synaptic changes in two groups, for a 'red' and a 'blue' case.

(see also below – text describes results in that same figure in a completely different way... )

Statement on faster learning with presynaptic plasticity is not well substantiated. Even in 3F – it depends how learning is defined; if it's defined as a separation of synaptic weights in early vs late groups, then building up of the separation will be about same (at least looks like in this resolution). It also might depend on temporal difference between early and late inputs.

For the correlation paradigm, 3I clearly shows that it's a matter of input parameters.

Which scenario is more 'typical' depends on which combination(s) of input frequency and correlation are more typical for actual neuronal activity.

l. 130-133. "This can be framed as a consequence of 130

potentiation requiring glutamate release [50], so that in a more reliable synapse, with 131

a high p value, there is a greater propensity for potentiation. Conversely, depression 132

was slower with presynaptically expressed plasticity, again because lowered probability 133

of release effectively also led to less plasticity"

Not clear what glutamate release has to do with plasticity in the model. As far as I understand, no mechanisms specific to glutamate release or glutamate receptors were implemented.

l. 143-145. "...in a scenario in 143

which there is competition due to e.g. limited resources, inputs with presynaptic 144

plasticity would be expected to overcome inputs with postsynaptic plasticity"

What if presynaptic resources are limited? e.g. Small ready-to-release pool or slow replenishment of vesicles?

l. 148 " Because of normalization, these two input populations competed, so 148

that one potentiated at the expense of the oher, which depressed."

First, what normalization? Methods do not say a word about it.

Second, how to reconcile "one population potentiated one depressed" with "which side potentiated faster" (Legend Fig 3I)? Slower potentiation is quite different from a depression.

l. 154 and further – "Presynaptic expression modelled as changes in short-term 154

plasticity".

Does this mean that 'presynaptic' model described so far did not include short-term plasticity? If yes, what is rationale for using such a model?

Fig. 4.

4E – what determined different steady-state of potentiated synapses in models with presynaptic (red) compared to postsynaptic or mixed (blue, black) expression?

4A-E, latency paradigm – would the rate of change (and the difference between models) depend on specifics of input patterns?

l. 176. "... synaptic efficacy was still potentiated faster and depressed 176

slower in the presynaptic case (Fig. 4E). " Not sure this is correct description. Red/early do potentiate more in the end, but rate of potentiation is about same as in black or blue, esp in the beginning.

Depression of 'late' looks pretty much the same in all three cases.

l. 185 In the simulations...

l. 198. "... in this model, LTP is expressed both pre- and postsynaptically, 198

whereas LTD is solely postsynaptically expressed."

l. 211. " postsynaptic plasticity in the tuned 211

model lacked the capacity to depress."

How the above two statements fit together?

and next, l. 214 "the inclusion of presynaptic LTD"...

Please explain clearly which model included what; and if 'biologically tuned' model does not include presynaptic LTD, why is it still included here?

l. 218. "groups of correlated and uncorrelated inputs 218

clustered (Fig. 6A) without the need for added competition through weight 219

normalization" .

Does this mean that simulations were run with and without weight normalization? Please clarify in the methods, how normalization was implemented, and how it was turned on and off. In Results and Figures, please clearly indicate which results were obtained with model(s) with normalization, and which in models without normalization.

l. 223-225 (and Fig 6C). Unclear, please explain.

l. 226. "The presynaptic frequency range for optimal 226

separation was between 50 and 80 Hz (Fig. 6C). At the other end of the range..."

What is 'the other end' of 50-80Hz range ???

l. 234. to to

l. 235,236 "In combination with postsynaptic plasticity, presynaptic plasticity provided a degree of output control..." - not clear, please explain.

Fig 5 Legend does not correspond to panel lettering.

l. 248 that that

l. 262. "Presynaptic plasticity expressed as the regulation of release 262

probability alone did not result in any differences over average postsynaptic activity 263

measurements compared to postsynaptic expression."

Fig 3 or related text do not show any data on average postsynaptic activity in correlation paradigm.

Methods.

Please define ALL parameters and terms in equations.

l.375-386. synaptic weight changes.

May be I am wrong (please correct/explain then), but unless you define qi= (0,1], qmax=Pmax =sqrt(qi) could be nonsense (say with qi=4).

l. 384. "The largest possible change for P or q separately was tot = 1 − qi."

This implies that q close to 1 limits changes in P; Why P can't change if qi=1?

l. 385-386 reads like bounds on P and q were set individually for different simulations; if so, please state this clearly and provide values of these limits for each simulation.

l. 405. "Presynaptic control of the probability of release per stimulus was implemented either 405

as a Markovian process or as short-term plasticity."

STP can be implemented using a Markovian process too, also at 5 release sites with initially uniform P. I am not suggesting the authors to do so; but if this should mean that no STP was implemented, it should be clearly stated.

**Have the authors made all data and (if applicable) computational code underlying the findings in their manuscript fully available?**

Reviewer #1: None

Reviewer #2: **No: **A link to the code still needs to be added

Reviewer #3: Yes

PLOS authors have the option to publish the peer review history of their article (what does this mean?). If published, this will include your full peer review and any attached files.

Reviewer #1: No

Reviewer #2: No

Reviewer #3: No
---

## [Decision Letter · Decision Letter 1]

14 Apr 2022

Dear Dr Sjöström,

Thank you very much for submitting your manuscript "Pre- and postsynaptically expressed spike-timing-dependent plasticity contribute differentially to neuronal learning" for consideration at PLOS Computational Biology. As with all papers reviewed by the journal, your manuscript was reviewed by members of the editorial board and by several independent reviewers. The reviewers appreciated the attention to an important topic. Based on the reviews, we are likely to accept this manuscript for publication, providing that you modify the manuscript according to the review recommendations.

Please take care to take into account the suggestions of reviewer#3 regarding careful proofreading of the text, in particular regarding its logical flow and consistency when referring to the literature.

Sincerely,

Hugues Berry

Associate Editor

PLOS Computational Biology

Kim Blackwell

Deputy Editor

PLOS Computational Biology

[LINK]

Reviewer's Responses to Questions

**Comments to the Authors:**

Reviewer #1: The paper is much clarified now.

Reviewer #2: The authors have addressed all of my concerns.

Reviewer #3: In the revised paper the authors addressed many of the concerns from previous comments.

However, I think a number of concerns remains not fully addressed.

Some parts of the text are still confusing (or have ambiguous meaning, implying what the authors probably did not mean to say), or plainly not correct.

Just few examples.

Introduction (which is by the way lengthy and lacks clear logics)

"In the 1990s, this led to a heated debate on the precise locus of expression of LTP, with 12

some arguing for postsynaptic expression, whereas others were in favour of a 13

presynaptic locus of LTP [8](Fig. 1A). Beginning in the early 2000’s, this controversy 14

was gradually resolved by the realisation that plasticity depends critically on several 15

factors, notably animal age, induction protocol, and precise brain region [9–11]. Indeed, 16

this resolution has now been developed to the point that it is currently widely accepted 17

that specific interneuron types have dramatically different forms of long-term 18

plasticity [12, 13], meaning that long-term plasticity in fact depends on the particular 19

synapse type [14]."

In the above, 1st sentence is about locus of expression;

2nd says controversy resolved, but talks about plasticity in general, without mentioning induction or expression or other features

3rd – all of a sudden talks about interneurons, while most of the debate on expression locus was about excitatory transmission to pyramidal neurons, mostly in the hippocampus...

Also, why interneurons where some peculiar forms of plasticity were reported in 2007 and later, and not a really classical paper from 1990 on pre and postsynaptic forms of plasticity in CA3 neurons (Zalutsky, Nicoll 1990), or clear demonstration of presynaptic mechanisms in neocortical neurons (Markram, Tsodyks 1995)?

I mean, the intro makes an impression that authors care about history of the problem.

"This is because presynaptically expressed plasticity leads 38

to changes in synaptic dynamics, whereas postsynaptic expression does not (Fig.1B). 39

For instance, during high-frequency bursting, as the readily releasable pool of vesicles in 40

a synaptic bouton runs out, leading to short-term depression of synaptic efficacy [29], 41

while short-term facilitation dominates at other synapse types [30]."

1st sentence – yes, OK.

2nd – this is not an elaboration of the statement in the previous sentence; also, readily releasable pool is also not infinite but is eventually depleted in synapses with PPF.

"As a corollary, it follows that presynaptic expression of 50

plasticity may change the computational properties of a given synaptic connection."

Why postsynaptic increase (or decrease) is NOT a change of computational property of a synapse? Yes, it does not change the filtering properties as presynaptic changes would, but it will change the probability of that input inducing a postsynaptic spike (well, on background of other synaptic activity) – why is this NOT a computational property of a synapse?

"In 51

this case, increasing the probability of release by the induction of LTP"

This implies that p can change only one way – increase, which is not true.

"Experimentally, it is long known that the induction of neocortical long-term 56

plasticity may alter short-term depression [16, 41]."

Can it also change short-term facilitation? Yes, many neocortical synapses show PPD, but some do show PPF (and in the hippocampus PPF is typical for many synapses).

I mean, the paper is not specifically about depressing, nor about specifically neocortical synapses, right?

(Discussion).

"Presynaptically expressed plasticity alone was not ideally suited for rate coding, 288

because it did not impact the average summed input effectively. "

This would be kind of true ONLY for the case of exclusively-bursting activity; It is definitely NOT true for the case of low-frequency spikes; also, in case of irregular firing changing p would change steady-state amplitude.

Also, plasticity and coding are two different 'dimensions'; plasticity can change coding, or mediate encoding of learned information, but plasticity is NOT coding;

"As a consequence, 289

postsynaptic firing frequency remained relatively unchanged after presynaptically 290

expressed plasticity (e.g., Figs. 4D, 5D, 6B)."

4D: about 15-20% increase of frequency, and about 40% decrease of burst duration; so kind of yes, total number of spikes in all bursts changed little.

This does not say much about firing frequency, calculated using all spikes.

5D: about 50% decrease of frequency in bursts, and about 60-80% decrease in burst duration;

-> about 5 – 10 fold (!) decrease of spikes in bursts. Unless there is a compensatory increase of single spike (I did not find such data in the Results), this does not look to me as "relatively unchanged".

6B – yes.

Thus, in the three groups of results, one example supports the claim, one does not, and one may be...

These are just some examples, there are many more in the text.

Bottom line: I would strongly recommend the senior authors to carefully read/edit the text, paying attention to preserving logical connections between sentences, overall consistency when referring to results of prior experimental and theoretical work, and consistency of interpretation of and conclusions from their own results.

**Have the authors made all data and (if applicable) computational code underlying the findings in their manuscript fully available?**

Reviewer #1: Yes

Reviewer #2: Yes

Reviewer #3: Yes

PLOS authors have the option to publish the peer review history of their article (what does this mean?). If published, this will include your full peer review and any attached files.

Reviewer #1: **Yes: **Paul Miller

Reviewer #2: **Yes: **Ian Stevenson

Reviewer #3: **Yes: **Maxim Volgushev

Figure Files:

Data Requirements:

Reproducibility:

References:

---

## [Decision Letter · Decision Letter 2]

11 May 2022

Dear Dr Sjöström,

We are pleased to inform you that your manuscript 'Pre- and postsynaptically expressed spike-timing-dependent plasticity contribute differentially to neuronal learning' has been provisionally accepted for publication in PLOS Computational Biology.

Best regards,

Hugues Berry

Associate Editor

PLOS Computational Biology

Kim Blackwell

Deputy Editor

PLOS Computational Biology

Reviewer's Responses to Questions

**Comments to the Authors:**

Reviewer #3: all my concerns addressed. no further comments.

**Have the authors made all data and (if applicable) computational code underlying the findings in their manuscript fully available?**

Reviewer #3: None

PLOS authors have the option to publish the peer review history of their article (what does this mean?). If published, this will include your full peer review and any attached files.

Reviewer #3: **Yes: **Maxim Volgushev

---

## [Editor Report · Acceptance letter]

1 Jun 2022

PCOMPBIOL-D-21-01572R2 

Pre- and postsynaptically expressed spike-timing-dependent plasticity contribute differentially to neuronal learning

Dear Dr Sjöström,

I am pleased to inform you that your manuscript has been formally accepted for publication in PLOS Computational Biology. Your manuscript is now with our production department and you will be notified of the publication date in due course.

With kind regards,

Anita Estes
